# Solvent-Free Coupling Reaction of Carbon Dioxide and Epoxides Catalyzed by Quaternary Ammonium Functionalized Schiff Base Metal Complexes under Mild Conditions

**DOI:** 10.3390/ma16041646

**Published:** 2023-02-16

**Authors:** Qin Wen, Xuexin Yuan, Qiqi Zhou, Hai-Jian Yang, Qingqing Jiang, Juncheng Hu, Cun-Yue Guo

**Affiliations:** 1Key Laboratory of Catalysis and Energy Materials Chemistry of Education & Hubei Key Laboratory of Catalysis and Materials Science, College of Chemistry and Materials Science, South-Central Minzu University, Wuhan 430074, China; 2School of Chemical Sciences, University of Chinese Academy of Sciences, Beijing 100049, China

**Keywords:** carbon dioxide, bifunctional metal complexes, cyclic carbonate, atmospheric pressure, activation energy

## Abstract

A series of bifunctional Schiff base metal catalysts (Zn-NPClR, Zn-NPXH, and M-NPClH) with two quaternary ammonium groups were prepared for carbon dioxide (CO_2_) and epoxide coupling reactions. The effects of the reaction variables on the catalytic activity were systematically investigated, and the optimal reaction conditions (120 °C, 1 MPa CO_2_, 3 h) were screened. The performances of different metal-centered catalysts were evaluated, and Co-NPClH showed excellent activity. This kind of bifunctional catalyst has a wide range of substrate applicability, excellent stability, and can be reused for more than five runs. A relatively high TOF could reach up to 1416 h^−1^ with Zn-NPClH as catalyst by adjusting reaction factors. In addition, the kinetic study of the coupling reaction catalyzed by three catalysts (Zn, Co, and Ni) was carried out to obtain the activation energy (*E*_a_) for the formation of cyclic carbonates. Finally, a possible mechanism for this cyclization reaction was proposed.

## 1. Introduction

Since the industrial revolution, large amounts of CO_2_ emissions are causing its concentration in the air to increase every year; it is now about 400 ppm (mg/L), leading to a series of environmental and social problems [1,2]. However, CO_2_ can be used as a non-toxic, cheap, easily available, and abundant C1 resource for constructing C-C, C-N, and C-O bonds to produce high-value chemicals such as carboxylic acids, oxazolidinones, urea, isocyanates, cyclic carbonates, etc. [3,4,5,6,7,8,9]. The synthesis of cyclic carbonates using carbon dioxide has been extensively studied. One is due to the 100% atomic utilization of the reaction, and the other is that the products have favorable physicochemical properties such as environmental friendliness, high boiling point, low odor, and low toxicity, and can be used as an alternative solvent to the high boiling point polar solvents DMF and DMSO [6]. In addition, cyclic carbonate can be applied to fuel additives, electrolytes for lithium-ion batteries [10,11], and polycyclic carbonate synthesis [12,13].

The thermodynamic and kinetic inertness of CO_2_ make its conversion require a large energy input (e.g., high temperature and pressure), and the search for suitable catalysts is a focus of research on whether CO_2_ can be used efficiently [6]. From a mechanistic point of view, in the synthesis of cyclic carbonates using CO_2_ and epoxides, catalysts have an important role in the ring opening of epoxides: (1) acting on the oxygen atom of the ring via Lewis acid or the formation of hydrogen bonds, and (2) nucleophilic reagents attacking the carbon atom of the epoxide to promote its ring opening [14]. Based on these presumptions, many catalytic systems have been developed for applications in the cycloaddition reactions of epoxides with carbon dioxide, such as ionic liquids [15,16,17,18], N-heterocyclic carbenes (NHCs) [19], porphyrins [20], organometallic framework materials (MOF) [21], inorganic catalysts [22], metal complex catalysts [23,24], carboxylic acid compounds [25,26,27], and quaternary ammonium salt-based catalysts [28,29].

Based on the need for catalytic systems to have both electrophilic and nucleophilic centers, various “bifunctional catalysts” have been developed [30,31,32]. In this paper, a variety of quaternary ammonium salt-based bifunctional metal catalysts were designed and synthesized for the cycloaddition reaction of epoxides with carbon dioxide (Figure 1). The results show that such bifunctional catalysts can effectively promote the formation of cyclic carbonate from epoxide and CO_2_ at low catalyst loadings (1 mol%) and relatively mild conditions (atmospheric pressure). In addition, since the reaction is heterogeneously catalyzed, catalyst recycling is simple and convenient. The catalyst recycling experiments and IR analysis of the catalysts showed that the prepared bifunctional catalysts can be effectively recycled five times without deactivation.

## 2. Materials and Methods

### 2.1. Chemicals and Analytical Methods

The information on materials is listed in Appendix A: Provenance and mass fraction purity of the materials. All the chemicals were used as received except for the epoxides, which were purified by distillation from CaH_2_ before utilization. The general procedure for the cycloaddition of CO_2_ to epoxides and spectra copies of all synthesized complexes and carbonate products are provided in the Appendix A. The yield and selectivity are determined by ^1^H NMR characterization.

### 2.2. Synthesis of Quaternary Ammonium Modified Metal Complexes Zn-NPClR, Zn-NPXH, M-NPClH

The synthetic route for catalysts is shown in Figure 2.

#### 2.2.1. Synthesis of 5-Chloromethyl Salicylaldehyde (PX, NPX)

The ligand precursors PX and NPX (X = Cl, Br, I) were prepared and characterized according to the method reported by Ji et al. (Figure 2) [33,34]. For comparison, a complex C_0_ was synthesized as a non-bifunctional catalyst.

#### 2.2.2. General Procedure for Synthesis of Quaternary Ammonium Modified Metal Complexes Zn-NPClR, Zn-NPXH

To a 100 mL dry round-bottom flask equipped with a magnetic stir bar, an ethanol solution (20 mL) of quaternary ammonium modified salicylaldehyde NPX (0.01186 mol) and an ethanol solution (10 mL) of 0.01533 mol amine C_6_H_6_R (R = H, Cl, NO_3_, CH_3_, C_4_H_9_) were added sequentially. The obtained mixture was heated and refluxed for 4 h. Then zinc acetate anhydrous (7.74 mmol) was added to the mixture. After another 6 h reflux, a large amount of precipitates were obtained. These precipitates were filtered and washed with ethanol several times, then dried in vacuum to obtain quaternary ammonium-modified salicylaldehyde-zinc catalysts. Details of these complexes are shown in Appendix A.

#### 2.2.3. Synthesis of Other Metal Center (Cobalt, Plumbum, Nickel, Copper) Complexes M-NPClH

Following the procedure of Zn-NPClH, we have also prepared different metal-centered quaternary ammonium salt complexes using other metal salts (cobalt chloride hexahydrate, lead acetate, nickel chloride hexahydrate, copper chloride dihydrate). The physicochemical information of these complexes is shown in Appendix A.

### 2.3. Catalytic Performance for CO_2_/Epoxide Cycloaddition Reaction

The cycloaddition reaction of CO_2_ and epoxide was conducted in a 50 mL stainless steel autoclave. The required catalyst and epoxide were added, and then CO_2_ was pressurized into the reactor at a certain pressure. The autoclave was immersed in an oil bath at a preset temperature, with stirring. After a proper time, the autoclave was cooled down to room temperature and then vented slowly. The resulting mixture was analyzed by ^1^H NMR to give the yield and the selectivity.

## 3. Results and Discussion

### 3.1. The Effect of Time on Cycloaddition of Propylene Oxide (PO) and CO_2_

To optimize the reaction conditions for the cyclization reaction of epoxide and carbon dioxide, Zn-NPClH was used as a catalyst and PO as a substrate. The effect of time on the reaction was investigated at 120 °C and 1 MPa CO_2_, as shown in Figure 1A. The propylene carbonate (PC) content increased steeply with time, reaching 3 h. It is obvious that within the initial 3 h, both the reaction rate and the PC yield increased rapidly. After 3 h, the PC yield still increased, but not as dramatically as before. The possible reason is that as the reaction proceeds, the PC concentration increases and the PO concentration decreases; in addition, the generated PC increases the viscosity of the system, which leads to a reduced opportunity for contact between the catalytically active site and the reaction substrate [35]. Therefore, 3 h was chosen as the optimal reaction time.

### 3.2. The Effect of Temperature on Cycloaddition of PO and CO_2_

The PC yield was low at 100 °C (Figure 1B), while increasing the temperature greatly improved the catalytic activity, which confirms the fact that the temperature has a significant effect on the reaction activity, and the fact that a high temperature can promote the PO conversion also indicates that this cycloaddition reaction is thermodynamically favorable [36,37,38]. When the temperature was increased from 100 °C to 120 °C, the PC yield increased steeply from 16% to 93%, but there was no significant change in the PC yield when the temperature was continued to increase to 140 °C (Figure 1B). This is because the cycloaddition reaction of CO_2_ and PO is exothermic, so too high a temperature will hinder the formation of cyclic carbonate. In addition, high temperatures will also lead to the polymerization of cyclic carbonate [39,40]. Based on the above study, 120 °C was chosen as the optimal reaction temperature.

### 3.3. The Effect of CO_2_ Pressure on Cycloaddition of PO and CO_2_

CO_2_ pressure affects the mass transfer kinetics of this cycloaddition reaction as well as the yield and selectivity of PC [41], so the relationship between pressure and yield was explored in the range of 0.5–5 MPa (Figure 1C). The yield of PC increased from 35% to 96%when the pressure was increased from 0.5 MPa to 1 MPa and decreased slightly to 94% when the pressure was continually increased to 2 MPa. Further raising the reaction pressure to 3 MPa and 5 MPa, the PC yield lowered to 73% and 66%, respectively. The increase in pressure makes the concentration of CO_2_ in PO higher, which promotes the reaction. However, too high pressure will reduce the concentration of PO in the vicinity of the catalyst, which will bring down the catalytic activity and thus reduce the reaction rate and the PC yield [42,43]. Experimental results show that the optimal CO_2_ pressure for this reaction is 1 MPa.

### 3.4. The Effect of the Ratio of PO to Catalyst on Cycloaddition of PO and CO_2_

Figure 1D shows the effect of catalyst loadings on PC yield at 120 °C and 1 MPa CO_2_. When the amount of catalyst was varied from 0.1 to 1 mol%, the PC yield dramatically improved from 10.2% to 93.2%. As the catalyst dosage increases, more catalytic active centers can be supplied to the catalytic system, so the PC yield is significantly increased [44]. However, further increasing the amount of the catalyst cannot lead to a big increase in the yield of carbonate; this may be due to the fact that the presence of excess catalyst makes its dispersion low in the system, thus hindering mass transfer between the catalyst active center and the reactants [45,46]. Therefore, the optimal ratio of catalyst to PO is 1:100.

### 3.5. Cycloaddition of CO_2_ to PO by Various Bifunctional Catalysts

The catalytic performances of various bifunctional catalysts were investigated for the cycloaddition reaction of PO and CO_2_ under optimal conditions (120 °C, 1 MPa, 3 h, 1:100). The corresponding results were summarized in Table 1. With the non-bifunctional complex C_0_ or Et_3_N as the sole catalyst, very low activity was observed (Table 1, entries 1,2). Using TBAC/Et_3_N or C_0_/Et_3_N as two-component catalysts, the catalytic activity was clearly improved (Table 1, entries 4,5). However, under the same conditions, a high yield of 93.2% was obtained using a quaternary ammonium salt bifunctional complex of Zn-NPClH as the catalyst (Table 1, entry 6). The electron effect of different substituents on ligands can affect the catalytic activity, so the effect of different ligands with electron-donating or electron-withdrawing substituents on the catalytic results was investigated. Under the same condition, the catalytic activity of the catalyst differs as follows: Zn-NPClNO_2_ > Zn-NPClCl > Zn-NPClH > Zn-NPClCH_3_ > Zn-NPClC_4_H_9_ (-NO_2_ > -Cl > -H > -CH_3_ > -C_4_H_9_) (Table 1, entries 6–10). This result demonstrates that the catalytic effect was related to the electronegativity of the substituents on the catalyst, and catalysts with a strong electron-withdrawing substituent showed better catalytic activity. It could be attributed to the ability of the strong withdrawing substituent, which can reduce the electron cloud density of the metal complexes and enhance their Lewis acidity. We also studied the effect of different metal centers in coordination compounds on catalytic activity (Table 1, entries 6,11–14). The results showed that cobalt, zinc, lead, and nickel metal center complexes have relatively good catalytic activity but poor activity for the copper center; the activity order is Co-NPClH > Zn-NPClH > Pb-NPClH > Ni-NPClH > Cu-NPClH (Co > Zn > Pb > Ni > Cu). The catalytic activity of different metal centers is highly dependent on the metal center type, which may be related to the different coordination abilities of metal centers with PO, while the low activity of bifunctional complexes with Cu metal centers may be attributed to the low acidity of Cu centers [47,48]. The anion plays an important role in the reaction of the epoxide with CO_2_ by nucleophilically attacking the epoxide to form a C-X bond to open its ring, so the anion species are screened. Accordingly, the more easily departed anion exhibits the best catalytic effect, which is consistent with the experimental results: I^−^ > Br^−^ > Cl^−^ (Table 1, entries6, 15, 16). To our delight, by adjusting reaction factors (0.1 mol% Zn-NPClH, 150 °C, 0.5 h), a relatively high TOF could reach up to 1416 h^−^^1^ with Zn-NPClH as catalyst (Table 1, entry 17).

### 3.6. Study of Catalytic Performance at Atmospheric Pressure

The catalytic activity of these bifunctional catalysts at atmospheric pressure has also been studied to meet the needs of sustainable development and green industrialization. The cycloaddition reaction of CO_2_ with isopropyl glycidyl ether was used as a model reaction to study the catalytic activity of the bifunctional metal catalysts M-NPClH at 0.1 MPa (Figure 2). All catalysts showed good to excellent activity. Overall, the order of catalytic activity of different metals at atmospheric pressure was consistent with that at high pressure: Co-NPClH > Zn-NPClH > Pb-NPClH > Ni-NPClH > Cu-NPClH. Zn-NPClH, Co-NPClH, Pb-NPClH, and Ni-NPClH all catalyzed glycidyl isopropyl ether better, with the catalyst Co-NPClH reaching 96.4% yield after 14 h of continuous reaction.

### 3.7. Substrate Scope of CO_2_/Epoxide Coupling

In order to investigate the generalizability of the bifunctional catalyst to other cycloaddition reactions of epoxides with CO_2_, we examined the adaptation of the bifunctional catalyst Zn-NPClH to a variety of epoxyalkane substrates under the above-mentioned optimized conditions, and the results are shown in Table 2 (Condition A). The results show that the catalyst system is effective in converting the studied common epoxides to their corresponding cyclic carbonates. For terminal epoxides, better yields of substrates with either electron-absorbing or electron-donating groups attached were obtained with the activation of the catalyst (Table 2, entries 1–2, 7–9). The nucleophilic attack of the epoxide by chloride ions is hindered by the high spatial site resistance of the epoxide environment, resulting in very low yields of dimethyloxirane, cyclohexene oxide, and 1,2-epoxyphenylethane (Table 2, entries 3–6), as previously reported in the literature [49,50,51]. In addition, the epoxide substrates of the glycidyl ether series can also be coupled with carbon dioxide, as in the case of dicyclic oxides, which can also be well formed as cyclic carbonates of the corresponding dicyclic ring. It is worth pointing out that bicyclic carbonates can be used as raw materials to produce non-isocyanate polyurethanes (NIPUs) by reacting with polyfunctional primary amines, which have important applications in industry [52,53].

The application of various epoxides with a higher boiling point (>110 °C) to couple with CO_2_ was also tested at atmospheric pressure. Although a longer time was necessary for the challenging epoxides (Table 2, Condition B), these bifunctional catalysts are generally still workable for a wide scope of epoxides, even under 0.1 MPa.

### 3.8. Catalyst Recycling

The reuse of catalysts in industry will not only reduce costs, but their reusability also reflects the stability of the catalyst under reaction conditions. Using Zn-PPBCl as a catalyst to catalyze the formation of PC from CO_2_ and PO as a model reaction, its reusability performance under optimal conditions was investigated. The synthesized Zn-PPBCl could be reused at least five times without significant loss of catalytic activity and selectivity (Figure 3). In addition, to further investigate the stability of this catalyst, the structures of both the5-cycleand fresh catalysts were also characterized by IR. As shown in Figure 4, the structure of the catalyst was maintained after five times of reuse, which proves that the synthesized catalyst is not only efficient but also stable and reusable.

### 3.9. Kinetic Studies

Kinetics study of the cycloaddition reaction using glycidyl isopropyl ether (GIE) as substrate using three bifunctional quaternary ammonium metal catalysts M-NPClH (M = Zn, Co, Ni) at 1 atm CO_2_ pressure (see the ESI, Appendix A: Structure of ligands–S15: Effects of the reaction time on cycloaddition of CO_2_ and glycidyl isopropyl ether at atmospheric pressure by Ni-NPClH at 393 K). The activation energies (*E*_a_) for the three catalysts were 30.8 kJ/mol (Zn), 26.1 kJ/mol (Co), and 32.9 kJ/mol (Ni), respectively, which were consistent with the order of catalytic activity (Figure 4, Figure 5 and Figure 6). The higher conversion obtained with Co-NPClH could likely be attributed to the higher Lewis acidity of the Co center compared with other metal centers [54].

## 4. Proposed Reaction Mechanism

Based on the experimental results, kinetic study analysis, and previously reported literature [53,54], we proposed a possible mechanism for the cycloaddition reaction of CO_2_ with epoxide (Figure 3). The metal center of the catalyst activates the oxygen on the epoxide, and the halogen ion on the functional group of the bifunctional metal complex catalyst nucleophilically attacks the less site-resistant carbon on the epoxide, causing the epoxide to undergo a ring-opening step to form a metal alcohol salt intermediate. At the same time, CO_2_ is inserted into this intermediate to form a metal carboxylate. Finally, the oxygen anion of the intermediate attacks the C-X bond, closing the loop to form propylene carbonate while completing the catalytic cycle to release the catalyst.

## 5. Conclusions

The coupling reactions of various epoxides with CO_2_ were successfully achieved by using halogenated quaternary ammonium salt metal complex catalysts without the addition of co-catalysts and solvents. When using Co-NPClH as a catalyst to promote the reaction with PO, the conversion was 98.6% and the PC selectivity was 99% under the optimal conditions (120 °C, 1 MPa CO_2_, and 1.0 mol% Co-NPClH). In addition, the synthesized bifunctional catalysts can be reused by simple operations and reused at least five times with essentially no decrease in activity or selectivity. The kinetic studies using catalysts M-NPClH (M = Co, Zn, Ni) are documented for the cycloaddition of GIE and CO_2_, in which the apparent activation energies (*E*_a_) are 26.1, 30.8, and 32.9 kJ/mol, respectively. The series of bifunctional metal catalysts synthesized in this study exhibited excellent catalytic performance, high TOF values, and strong stability, which proved the potential of these catalysts for industrial applications.

## Data Availability

The data presented in this study are openly available in MDPI.

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
