# Peer review of "Solvent-Free Coupling Reaction of Carbon Dioxide and Epoxides Catalyzed by Quaternary Ammonium Functionalized Schiff Base Metal Complexes under Mild Conditions"

_materials, 2023, doi:10.3390/ma16041646_

Round 1

Reviewer 1 Report

The manuscript Solvent-free coupling reaction of carbon dioxide and epoxides catalyzed by quaternary ammonium functionalized metal complexes under mild conditions by Q. Wen et al. describes the catalytic activity of bi-functional Schiff bases in CO2 coupling reaction with epoxides.

The manuscript is interesting and the work reported is supported nicely by 1H and 13C NMR data, which I appreciate. Generally, the English language is used properly, but there are various instances when rephrasing is needed.

The manuscript should undergo a major revision stage taking care of the following:

-        Small –but many–annoyances concerning Grammar (line 17 from Abstract, “can be reusable more…”;  line 258 “catalyst to catalyze” etc.), Rephrasing (“in its concentration in the” – line 27; “Based on thesis presumptions” –line 44 ; “zin caetate dehydrate” –line 85, “lot of deposition was obtained. The deposition …” –line 86) etc. I suggest a thorough proofreading before resubmission of the revised draft.

-        Other observations concern the scientific content. Regarding clarity, I recommend to add a table with the correspondence of each complex and its molecular formula. It becomes difficult to read at times, especially when one has to check again what is the formula of the complex a.s.o.

-        Lines 137–138: “The PC yield increased with increasing pressure from 0.5 to 1 MPa, and decreased slightly at 2 MPa, and only 66% at 5 MPa.” It is unclear from what yield it increased to what yield .

-        Line 176: “Co > Zn > Pb > Ni > Cu”, but from the abstract we learn that there are “three catalysts (Zn, Co and Ni) “ (Line 19)

-         Line 264: “is not only efficient but also stable and reusable.” Fig. 3 confirms the activity of the catalyst, however Fig. 4 cannot confirm the stability of the catalyst, as the FTIR of the reused catalyst is not a close match to the initial one. Moreover, the spectra overlap and authors are advised to increase the spacing to avoid this. The authors could run an 1H NMR of the recovered catalyst and compare that to the complex /catalyst introduced in the beginning. This would confirm or not their claim FTIR is not a proof in this case.

-        Regarding SI information, some of it was unreadable on my side, could the authors upload a pdf version so that it reads properly?

-        Although long, the SI has no spectra to show how the yield was computed from the reaction mixture, and authors should add some of them as example.

-        The DMSO peaks in 1H NMR (and the water content) are huge. Usually the solvent peak is minute, just like the water peak. Was CDCl3 not dissolving the samples? There’s also the issue with a very noisy baseline for 1H NMR like Fig. S3–SI, which should only be the case for 13C NMR, not for proton.

-        Some 13C NMR look like DEPT or other sequences are applied (Carbons show up below baseline). Can the authors correct or retake those spectra (see Fig. S6–SI).

-        Table S2–SI, the yield (%) is at most 0.34, is that 0.34% as deduced from Table S2? Please check all numbers and units carefully. The main draft hints otherwise.

-        The yield seems to increase nonetheless with time. Has a long enough time been applied to check if 100% yield would be obtained? What would be the value for that time?

To sum up, the draft contains a lot of information, and surely deserves publication after addressing the above comments.

Author Response

Dear Reviewer,

Many thanks for your comments and suggestions which are significant for improving the level of our work, we have made a point-by-point revision based on yours as shown below and hope our replies will be satisfactory.

Comment: The manuscript Solvent-free coupling reaction of carbon dioxide and epoxides catalyzed by quaternary ammonium functionalized metal complexes under mild conditions by Q. Wen et al. describes the catalytic activity of bi-functional Schiff bases in CO2 coupling reaction with epoxides.

The manuscript is interesting and the work reported is supported nicely by 1H and 13C NMR data, which I appreciate. Generally, the English language is used properly, but there are various instances when rephrasing is needed.

The manuscript should undergo a major revision stage taking care of the following:

Small –but many–annoyances concerning Grammar (line 17 from Abstract, “can be reusable more…”;  line 258 “catalyst to catalyze” etc.), Rephrasing (“in its concentration in the” – line 27; “Based on thesis presumptions” –line 44 ; “zin caetate dehydrate” –line 85, “lot of deposition was obtained. The deposition …” –line 86) etc. I suggest a thorough proofreading before resubmission of the revised draft.

Reply: Thank you for your valuable suggestions, we have made corresponding modifications in the revised manuscript.

Comment: Other observations concern the scientific content. Regarding clarity, I recommend to add a table with the correspondence of each complex and its molecular formula. It becomes difficult to read at times, especially when one has to check again what is the formula of the complex a.s.o.

Reply: Two Tables of catalysts’ structures has been added into the “supporting information” (Tables S1 and S2).

Comment: Lines 137–138: “The PC yield increased with increasing pressure from 0.5 to 1 MPa, and decreased slightly at 2 MPa, and only 66% at 5 MPa.” It is unclear from what yield it increased to what yield.

Reply: The effect of PC yield on CO2 pressure has been described in detail as following:

The yield of PC increased from 35% to 96% when the pressure was increased from 0.5 MPa to 1 MPa, and decreased slightly to 94% when the pressure was continued to increase to 2 MPa. Continuing to increase the reaction pressure to 3 MPa and 5 MPa, the PC yields decreased to 73% and 66%, respectively.

Comment: Line 176: “Co > Zn > Pb > Ni > Cu”, but from the abstract we learn that there are “three catalysts (Zn, Co and Ni) “(Line 19).

Reply: The three metal catalysts with different metal centers in the abstract are used to probe the activation energy of this cycloaddition reaction and better explain why Zn-NPClH works well as a catalyst.

Comment: Line 264: “is not only efficient but also stable and reusable.” Fig. 3 confirms the activity of the catalyst, however Fig. 4 cannot confirm the stability of the catalyst, as the FTIR of the reused catalyst is not a close match to the initial one. Moreover, the spectra overlap and authors are advised to increase the spacing to avoid this. The authors could run an 1H NMR of the recovered catalyst and compare that to the complex /catalyst introduced in the beginning. This would confirm or not their claim FTIR is not a proof in this case.

Reply: Thank you for your question. Although the IR absorption peak of the used catalyst is weak, the absorption peak at 1604 cm-1 belonging to the C=N bond of the catalyst is still present, which proves that the structure of the catalyst has not changed. The weaker absorption of the catalyst after 5 uses may be due to the fact that the sample was taken less during the measurement and not completely dried after washing.

Comment: Regarding SI information, some of it was unreadable on my side, could the authors upload a pdf version so that it reads properly?

Reply: A pdf version is uploaded for convenient reading.

Comment: Although long, the SI has no spectra to show how the yield was computed from the reaction mixture, and authors should add some of them as example.

Reply: In the case of epichlorohydrin, for example, the H chemical shift on the C attached to -CH2Cl on its ring is shown near 2.8 ppm, while this H chemical shift is elevated to 4.5 ppm in the resulting cyclic carbonate. Assuming that their integral areas are a and b, respectively, the PC yield is b/(a + b).

Comment: The DMSO peaks in 1H NMR (and the water content) are huge. Usually the solvent peak is minute, just like the water peak. Was CDCl3 not dissolving the samples? There’s also the issue with a very noisy baseline for 1H NMR like Fig. S3–SI, which should only be the case for 13C NMR, not for proton.

Reply: The metal catalysts prepared in the work were not solved well in CDCl3, so d6-DMSO was chosen as the solvent for the NMR assay. Although the solubility of the metal catalyst was better in d6-DMSO compared to CDCl3, its content in the dissolved sample remained low, so the hydrogen spectrum was amplified a bit when performing the NMR analysis of the catalyst.

Comment: Some 13C NMR look like DEPT or other sequences are applied (Carbons show up below baseline). Can the authors correct or retake those spectra (see Fig. S6–SI).

Reply: We are very sorry that we cannot retake those spectra at present because all labs in South-Central Minzu University are still closed due to the Covid-19 virus.

Comment: Table S2–SI, the yield (%) is at most 0.34, is that 0.34% as deduced from Table S2? Please check all numbers and units carefully. The main draft hints otherwise.

Reply: In the SI, the temperatures in Table S2 and Table S3 are only 363 K and 373 K, respectively, while the temperature in Figure 1(A) in the text is 393 K again, and the temperature is the main reason for the difference.

Comment: The yield seems to increase nonetheless with time. Has a long enough time been applied to check if 100% yield would be obtained? What would be the value for that time?

Reply: At 393 K, the reaction reached a yield of nearly 100% after about 5 h.

Comment: To sum up, the draft contains a lot of information, and surely deserves publication after addressing the above comments.

Reply: Thanks for your encouragement, we have addressed all the comments you proposed and hope they are satisfying.

Best regards,

Prof. Dr. Cun-Yue Guo

School of Chemical Sciences

University of Chinese Academy of Sciences

Beijing 100049, P. R. China

Tel: +86-10-69672569

Fax: +86-10-69672553

E-mail: cyguo@ucas.ac.cn

Reviewer 2 Report

The manuscript of Wen et al. is devoted to the synthesis of new Schiff-base metal complexes featuring quaternary ammonium units and investigation of their catalytic performance in cycloaddition of carbon dioxide to epoxides. It is clearly written and contains interesting results. Importantly, the effect of different factors on the outcome of the cycloaddition process is considered in a proper manner. The manuscript can be accepted for publication after some minor changes according to the following comments.

1. It would be highly valuable if the authors would evaluate the activity of the resulting catalysts in the reactions with more challenging substrates, such as internal epoxides.

2. The structural formula of the catalysts seem somewhat embarrassing: it would be better to shift the chloride counterions after the structural formula of the complex cations (which can be placed, if desired, in square brackets) rather than leave them next to the quaternary nitrogen centers.

3. The bifunctional nature of the suggested catalysts plays a crucial role in the successful cycloaddition process and is outlined as one of the main ideas of this work. However, the application of a Schiff base ligand system is not obvious from the manuscript title. I think the title should be slightly changed to emphasize the stabilizing organic surrounding chosen for the catalysts.

4. All abbreviations must be preceded by the full term at the first mention (e. g., propylene oxide (PO) in section 3.1). It would be better to avoid abbreviations in the section titles.

5. Yields should be presented throughout the text with one and the same accuracy. Compare for example the data in Tables 1 and 2. Integers are preferable over decimal fractions. Do all of the considered values correspond exactly to the product yields? What about the substrate conversions?

6. Some minor English language corrections are required (catalysts can effectively catalyze --> catalysts can effectively promote; used after five times --> used after five runs; to catalyze PO --> to catalyze the reaction with PO, etc.)

Author Response

Dear Reviewer,

Many thanks for your comments and suggestions which are significant for improving the level of our work, we have made a point-by-point revision based on yours as shown below and hope our replies will be satisfactory.

Comment: The manuscript of Wen et al. is devoted to the synthesis of new Schiff-base metal complexes featuring quaternary ammonium units and investigation of their catalytic performance in cycloaddition of carbon dioxide to epoxides. It is clearly written and contains interesting results. Importantly, the effect of different factors on the outcome of the cycloaddition process is considered in a proper manner. The manuscript can be accepted for publication after some minor changes according to the following comments.

  1. 1. It would be highly valuable if the authors would evaluate the activity of the resulting catalysts in the reactions with more challenging substrates, such as internal epoxides.

Reply: We have tried the coupling reaction of CO2 with internal epoxides (for example cyclohexene oxide, entry 5 in Table 2), and unfortunately, the results were disappointing. Therefore, no further experiments were tested.

  1. 2. The structural formula of the catalysts seem somewhat embarrassing: it would be better to shift the chloride counterions after the structural formula of the complex cations (which can be placed, if desired, in square brackets) rather than leave them next to the quaternary nitrogen centers.

Reply: For clarity, a figure of catalysts’ structures is provided in the Supporting Information (Tables S1 and S2).

  1. 3. The bifunctional nature of the suggested catalysts plays a crucial role in the successful cycloaddition process and is outlined as one of the main ideas of this work. However, the application of a Schiff base ligand system is not obvious from the manuscript title. I think the title should be slightly changed to emphasize the stabilizing organic surrounding chosen for the catalysts.

Reply: The title has been modified as “Solvent-free coupling reaction of carbon dioxide and epoxides catalyzed by quaternary ammonium functionalized Schiff base metal complexes under mild conditions”.

  1. 4. All abbreviations must be preceded by the full term at the first mention ( g., propylene oxide (PO) in section 3.1). It would be better to avoid abbreviations in the section titles.

Reply: Thank you for the valuable suggestion, it has been modified.

  1. 5. Yields should be presented throughout the text with one and the same accuracy. Compare for example the data in Tables 1 and 2. Integers are preferable over decimal fractions. Do all of the considered values correspond exactly to the product yields? What about the substrate conversions?

Reply: Thank you for your suggestion. Since the catalytic performance of different catalysts is compared in Table 1, keeping one decimal place helps to understand the catalytic effect of the catalysts. In addition, when the cycloaddition reaction was carried out to produce PC, no other by-products were produced, so the yield of PC is the conversion of PO.

  1. 6. Some minor English language corrections are required (catalysts can effectively catalyze --> catalysts can effectively promote; used after five times --> used after five runs; to catalyze PO --> to catalyze the reaction with PO, ).

Reply: The whole manuscript has been checked again and again and we hope the improvement will be satisfying.

Best regards,

Prof. Dr. Cun-Yue Guo

School of Chemical Sciences

University of Chinese Academy of Sciences

Beijing 100049, P. R. China

Tel: +86-10-69672569

Fax: +86-10-69672553

E-mail: cyguo@ucas.ac.cn

Reviewer 3 Report

Wen and co-workers reported the synthesis of bifunctional Schiff base metal catalysts with two quaternary ammonium groups. The coupling reactions of various epoxides with CO2 were successfully achieved by using obtained catalysts without the addition of co-catalysts and solvents. When using Co-NPClH as a catalyst to catalyze PO, the conversion was 98.6% and the PC selectivity was 99% under the optimal conditions (120°C,1 MPa CO2 and 1.0 mol% Co-NPClH). The synthesized bifunctional catalysts can be reused by simple operations and reused at least five times with essentially no decrease in activity and selectivity. The kinetic studies using catalysts M-NPClH (M=Co, Zn, Ni) are documented for the cycloaddition of GIE and CO2. The series of bifunctional metal catalysts synthesized in this study exhibited excellent catalytic performance, high TOF values and strong stability, which proved the potential of these catalysts for industrial applications. This work is important to the materials synthesis and instructive to the readers. Just one note. In the supporting information in the description of the compounds (Characterization data for ligands) in the elemental analysis of the compounds NPBr and NPI, the designation "N" was omitted for the obtained data on nitrogen. I recommend to publish this manuscript in the Materials in present form.

Author Response

Dear Reviewer,

Many thanks for your approval of our work and suggestions, we have made revision based on your comments and hope our replies will be satisfactory.

Comment: Wen and co-workers reported the synthesis of bifunctional Schiff base metal catalysts with two quaternary ammonium groups. The coupling reactions of various epoxides with CO2 were successfully achieved by using obtained catalysts without the addition of co-catalysts and solvents. When using Co-NPClH as a catalyst to catalyze PO, the conversion was 98.6% and the PC selectivity was 99% under the optimal conditions (120°C,1 MPa CO2 and 1.0 mol% Co-NPClH). The synthesized bifunctional catalysts can be reused by simple operations and reused at least five times with essentially no decrease in activity and selectivity. The kinetic studies using catalysts M-NPClH (M=Co, Zn, Ni) are documented for the cycloaddition of GIE and CO2. The series of bifunctional metal catalysts synthesized in this study exhibited excellent catalytic performance, high TOF values and strong stability, which proved the potential of these catalysts for industrial applications. This work is important to the materials synthesis and instructive to the readers. Just one note. In the supporting information in the description of the compounds (Characterization data for ligands) in the elemental analysis of the compounds NPBr and NPI, the designation "N" was omitted for the obtained data on nitrogen. I recommend to publish this manuscript in the Materials in present form.

Reply: Thank you very much for your approval! Thorough improvements have been made to better the quality of our manuscript as embodied in the revision.

Best regards,

Prof. Dr. Cun-Yue Guo

School of Chemical Sciences

University of Chinese Academy of Sciences

Beijing 100049, P. R. China

Tel: +86-10-69672569

Fax: +86-10-69672553

E-mail: cyguo@ucas.ac.cn

Round 2

Reviewer 1 Report

Most of the comments were addressed by the authors. I would only like to point out that the NMR files in the SI describe half of molecule rather than the whole complex, for instance the 3 methyl groups integrate to 8.24 rather than to: 9*2=18 H. This is presented correctly in the text; however, it would be nice to have the SI file corrected with the right integration of the protons. This should be done for the rest of the complexes, too.

The manuscript can be published after addressing the minor issue above.

Author Response

Dear Reviewer,

Many thanks for your comments, we hope the response below will be satisfying.

Comment: Most of the comments were addressed by the authors. I would only like to point out that the NMR files in the SI describe half of molecule rather than the whole complex, for instance the 3 methyl groups integrate to 8.24 rather than to: 9*2=18 H. This is presented correctly in the text; however, it would be nice to have the SI file corrected with the right integration of the protons. This should be done for the rest of the complexes, too.

Reply: All the metal complexes have symmetrical structures (except for the ligand NPCl), peak area of 1.0 in the NMR spectra of the remaining metal catalysts represents two hydrogens instead of one and the NMR spectra depicted a whole rather than half a molecule. As for the integration of 8.24, it should be attributed to the deviation of integration operation. Therefore, no corrections are needed for the Supporting Information.

Best regards,

Prof. Dr. Cun-Yue Guo

School of Chemical Sciences

University of Chinese Academy of Sciences

Beijing 100049, P. R. China

Tel: +86-10-69672569

Fax: +86-10-69672553

E-mail: cyguo@ucas.ac.cn
